# Early Summer Temperature Variation Recorded by Earlywood Width in the Northern Boundary of *Pinus taiwanensis* Hayata in Central China and Its Linkages to the Indian and Pacific Oceans

**DOI:** 10.3390/biology11071077

**Published:** 2022-07-19

**Authors:** Meng Peng, Xuan Li, Jianfeng Peng, Jiayue Cui, Jingru Li, Yafei Wei, Xiaoxu Wei, Jinkuan Li

**Affiliations:** 1College of Geography and Environmental Science, Henan University, Kaifeng 475004, China; pengmeng@henu.edu.cn (M.P.); lixuan@vip.henu.edu.cn (X.L.); cuijiayue@henu.edu.cn (J.C.); lijingru@vip.henu.edu.cn (J.L.); weixu@henu.edu.cn (X.W.); 104753210170@henu.edu.cn (J.L.); 2The Key Laboratory of Earth System Observation and Simulation of Henan Province, Kaifeng 475004, China; 3National Demonstration Center for Environment and Planning, Henan University, Kaifeng 475004, China

**Keywords:** Tongbai Mountains, *Pinus taiwanensis* Hayata, tree-ring width, temperature variation, SST

## Abstract

**Simple Summary:**

This paper analyzed the different relationships between earlywood and latewood as well as total tree-ring growth and the climate factors and reconstructed 106 years of May–June mean temperature (T_MJ_) in the Tongbai Mountains based on the earlywood width chronology of *Pinus taiwanensis* Hayata. It also analyzed the linkages to the Indian and Pacific Oceans. This paper found that earlywood width chronology has better response to the climate factors than latewood width and total tree-ring width. This study also found that the main limiting factors that restrained radial growth of *Pinus taiwanensis* Hayata in the Tongbai Mountains were May–June mean temperature and mean maximum temperature. The reconstructed T_MJ_ series have a better reliability and are significantly negatively correlated with sea surface temperature (SST) over the tropical Western Pacific Ocean and Indian Ocean and are significantly positively correlated with SST over the subtropical Pacific Ocean. Finally, the periodic fluctuations of T_MJ_ in the Tongbai Mountains might be related to the quasi-biennial interannual oscillation of SST over the Indo-Pacific equatorial region (QBO). The results of this study are significant for further understanding and exploring forest growth and climate change in the climatic transition zone.

**Abstract:**

The Tongbai Mountains are an ecologically sensitive region to climate change, where there lies a climatic transitional zone from a subtropical to a warm–temperate monsoon climate. The northern boundary of *Pinus taiwanensis* Hayata is here; thus, climate information is well recorded in its tree rings. Based on developed earlywood width (EWW), latewood width (LWW) and total ring width (RW) chronologies (time period: 1887–2014 year) of *Pinus taiwanensis* Hayata in the Tongbai Mountains in central China, this paper analyzed characteristics of these chronologies and correlations between these chronologies and climate factors. The correlation results showed that earlywood width chronology contains more climate information than latewood width chronology and total ring width chronology, and mean temperature and mean maximum temperature in May–June were the main limiting factors for radial growth of *Pinus taiwanensis* Hayata. The highest significant value in all correlation analyses is −0.669 (*p* < 0.05) between earlywood width chronology and May–June mean temperature (T_MJ_) in the pre-mutation period (1958–2005) based on mutating in 2006. Thus, this paper reconstructed May–June mean temperature using earlywood width chronology from 1901 to 2005 (reliable period of earlywood width chronology is 1901–2014). The reconstructed May–June mean temperature experienced eight warmer periods and eight colder periods and also showed 2–3a cycle change over the past 105 years. The spatial correlation showed that the reconstructed series was representative of the May–June mean temperature variation in central and eastern China and significant positive/negative correlation with the sea surface temperature (SST) of the subtropical Pacific Ocean and the tropical Western Pacific Ocean and Indian Ocean from the previous October to the current June. This also indicated that May–June mean temperature periodic fluctuations might be related to the quasi-biennial oscillation (QBO) in the tropical Western Pacific Ocean and Indian Ocean. The results of this study have extended and supplemented the meteorological records of the Tongbai Mountains and have a guiding significance for forest tending and management in this area.

## 1. Introduction

The sixth IPCC research report showed that the global mean surface temperature has risen about 1 °C since 1900 and is expected to reach or exceed 1.5 °C from the perspective of mean temperature change in the next 20 years; thus, global warming had become fact [1]. Global warming causes changes in the structure and function of forest ecosystems and trees, and previous studies showed that tree xylem growth is controlled by climate factors, and climate change could lead to changes in tree-ring width by affecting the process of tree physiology and growth; hence, the data of tree rings can better record and reflect the current climate change information [2,3]. In order to improve the ability of predicting global climate change in the future, the PAGES (Past Global Changes) program proposed to study past climate change by using different proxy indicators (including tree ring, pollen, ice core, loess, lake sediment, coral, etc.) [4,5]. Tree rings have become an important proxy of studying past climate change at the millennial scale based on high resolution, accurate dating and wide distribution [6].

Tree-ring studies in China have developed rapidly in recent years. In addition to the traditional tree-ring studies in the Tibet Plateau and the northwestern arid area in China [7], tree-ring research in the eastern humid area in China has consecutively developed rapidly. For example, the studies include the response of tree-ring radial growth to climate change [8,9,10,11,12,13,14,15,16,17], and the reconstruction of climate factors [18,19,20,21,22,23,24,25,26,27,28,29,30,31,32] was from the Greater Khingan Mountain, Mt. Changbai, Mt. Taihang, Mt. Qinling and Mt. Funiu. Likewise, there are many studies in the southeast of China that analyze the relationship between tree growth and climate [33,34,35,36,37] and past climate change reconstruction [38,39,40] based on the tree rings of *Pinus massoniana* Lamb. and *Pinus taiwanensis* Hayata.

The process of tree-ring formation involves cambial cell division, cell enlargement, cell wall thickening, and finally, xylem cell maturation [2]. Earlywood is the segment that is formed by cell division enlargement in the beginning of the growing season and is characterized by a thin cell wall, large size and light color. Latewood is the segment that is formed by cell division in the late growing season and is characterized by a thick cell wall, small size and dark color [41]. Compared with RW, the EWW and LWW are more sensitive to climate factors and provide higher resolution [42]. Therefore, dendrologists are interested in analyzing the response between tree growth and climate factors and reconstructing past climate change based on EWW and LWW. Stahle et al. [43] reconstructed the winter–spring and summer precipitation variations in northwestern New Mexico based on EWW and LWW of *Pseudotsuga menziesii* and *Pinus ponderosae*. Therrell et al. [44] pointed out that LWW of *Pseudotsuga menziesii* in northern Mexico was sensitive to precipitation from June to August, but LWW of *Pseudotsuga menziesii* in southern Mexico was sensitive to precipitation from April to June. Griffin et al. [45] showed that LWW of *Pseudotsuga menziesii* in the southwestern United States was sensitive to precipitation and could be used to reconstruct precipitation during the rainy season of North America. The tree-ring studies of EWW and LWW have also been carried out in China in recent years. Zhang et al. [46] found that EWW and LWW and RW of *Picea meyeri* Rehd. et Wils. were sensitive to temperature in the Hunshandake Sandy Land. Zhao et al. [47] found that EWW of *Cryptomeria japonica* (L. f.) D. Don in the area of southwestern Sichuan was sensitive to temperature, and temperature had a “lag effect” on EWW growth. Zhao et al. [48] found that EWW of *Pinus tabulaeformis* Carr. in the eastern Qinling Mountains was more sensitive to the climate factors than LWW and RW and reconstructed self-calibrated Palmer Drought Severity Index of this area based on EWW chronology. Feng et al. [49] showed that EWW of *Pinus armandi* Franch. in the Huashan Mountains was greatly affected by a mean daily temperature of 3 °C, while LW was greatly affected by a mean daily temperature of 8 °C, and it would limit the growth of trees when the temperature was higher than 11 °C. Gu et al. [50] pointed out that EWW was more sensitive to the climate than LWW based on the margin and central distribution area of *Pinus massoniana* Lamb. Zhao et al. [51] found that LWW growth of *Tsuga longibracteata* Cheng trees in the border area of Hunan and Guangxi provinces was significantly correlated with precipitation and temperature, and they reconstructed a variation of the SPEI index in this region based on LWW chronology. Ma et al. [52] found the EWW and LWW of *Pinus taiwanensis* Hayata in the Sanqing Mountains, which showed a consistent “lag effect” on climate factors.

The Tongbai Mountains and Dabie Mountains are both located in a climatic transition region from the north subtropics to the warm–temperate climate, and the more tree-ring research in the Dabie Mountains was achieved, mainly including dendroecology and dendroclimatology based on *Pinus taiwanensis* Hayata and *Pinus tabulaeformis* Carr. as well as *Pinus massoniana* Lamb. [53,54,55,56,57,58,59,60,61]. Tongbai Mountains is western and northern distribution boundary of *Pinus taiwanensis* Hayata and northern distribution boundary of *Pinus massoniana* Lamb. in mainland China and these trees are sensitive to climate change. However, there are only Cai and Liu [62] analyzes the growth of *Pinus massoniana* Lamb. and reconstructs the variation of May–July mean minimum temperature. Therefore, studies EWW and LWW of *Pinus taiwanensis* Hayata in the Tongbai Mountains are helpful to further explore the relationship between tree growth and climate factors at a higher temporal resolution and to reconstruct past climate variations. This study aims to: (1) mainly determine the limiting climatic factors on the growth of EWW, LWW and RW of *Pinus taiwanensis* Hayata, (2) reconstruct and analyze climate change by EWW or LWW or RW of *Pinus taiwanensis* Hayata, and (3) explore spatial representatives and the driving mechanisms of climate change in the region.

## 2. Materials and Methods

### 2.1. Study Area

As shown in Figure 1, the Tongbai Mountains are located in the junction between Tongbai county in Henan Province and Suizhou county in Hubei Province. The belong to the western section of the Huaiyang Mountains, running northwest to southeast and connecting the eastern Mt. Funiu to the western Dabie Mountains. They are more than 120 km long and Taibai peak is 1140 m above sea level, the birthplace of the Huai River. Moreover, the Tongbai Mountains are located in the transition zone from the subtropical to the warm–temperate monsoon humid climate, and they hold abundant rain and distinct seasons, with a mean annual temperature of 16.6 °C and a mean annual precipitation of 1110 mm. The basal zone soil in the north slope of the Tongbai Mountains is yellow-brown soil, which is mainly distributed in the low mountains and hilly areas at altitudes of less than 700 m. The dark yellow-brown soil is mainly distributed in the areas with an altitude of 700–1000 m, which belongs to the transition type from yellow-brown soil to brown soil in the spectrum of soil vertical distribution. The mountain brown soil is mainly distributed in the areas with an altitude of more than 1000 m [63]. The vegetation in the Tongbai Mountains is characterized by a north–south transition, which is a mixed evergreen coniferous broad-leaved forest and deciduous broad-leaved forest. There are many types of vegetation in the Tongbai Mountains, such as pine forests, Quercus forests and bushes, which include *Pinus taiwanensis* Hayata, *Pinus massoniana* Lamb., Pinus tabulaeformis Carr., *Quercus variabilis* Blume, *Quercus glauca* Thunb., *Quercus acutissima* Carruth., *Cotinus coggygria* Scop., *Forsythia suspensa* (Thunb.) Vahl, *Rhododendron simsii* Planch, etc. [64]. Because the Tongbai Mountains are located in the climate transition zone and because the northern and western distribution boundary of *Pinus taiwanensis* Hayata is in mainland China [55], the natural forest of *Pinus taiwanensis* Hayata in the Tongbai Mountains is an ideal place to study the tree specie’s relationship between tree rings and climate factors.

### 2.2. Tree Ring Data

As shown in Figure 1, in May 2015, we collected the tree-ring data of *Pinus taiwanensis* Hayata from the natural forest (32°24′ N, 113°16′ E, 563 m a.s.l.) of the Tongbai Mountains, labeled THD01. According to the basic principles and criteria of tree-ring research [65], 2 cores were taken from each tree, and 50 cores were collected from 25 trees in total. 

The samples were brought back to the laboratory and fixed and polished according to the standard methods of the international tree-ring database [66]. After pretreatment, the preliminary dating was carried out. The boundary of EW and LW was clear and easy to distinguish in most cases, but it was difficult to distinguish the boundary of the transition between EW and LW in a few sample cores; thus, this paper determined the standard boundary of EW and LW based on the method of Stahle et al. [43]. As shown in Figure 2, which was captured by an Olympus CX 43 light microscope (200× magnification), (1) we defined the mutant line as the boundary between EW and LW where there was a mutation from big, light cells to small, dark cells; (2) we defined the middle line of the gradual change area as the boundary where there was a gradual change between EW and LW. 

Based on the above-mentioned criteria, the MeasureJ2X tree-ring width measurement system (precision 0.001 mm) and the Velmex tree-ring width instrument, the width of the earlywood and latewood was measured per core year by year, and three width sequences of earlywood, latewood and total tree-ring were obtained. The COFECHA program [67] was used to inspect the quality of the three tree-ring width sequences, and then we removed the cores with short tree age, decay and unclear boundaries between earlywood and latewood. Finally, we selected the sequence values of 39 cores from 20 trees. The ARSTAN program [68] was used to fit the series of EWW, LWW and RW by linear function or by negative exponential function, which allowed us to remove the trees growth trend and correct the growth amount. Finally, we developed the standard chronologies (STD) and residual chronologies (RES) of EWW, LWW and RW. The standard chronology removes the trees’ age-related growth trend, while the residual chronology removes not only the trees’ age-related growth trend but also the persistent influence of individual trees on later growth due to local microenvironmental changes.

### 2.3. Climate Data

The meteorological data of the Tongbai meteorological station (32°23′ N, 113°25′ E, 1958–2014, 153 m a.s.l.), which is about 15 km from the sample site, were selected as reference from the China Meteorological Data Service Center (http://data.cma.cn/, accessed on 1 May 2021), as shown in Figure 3. Considering the lag effect of climate factors on tree growth, this paper selects the monthly total precipitation (P) and monthly mean temperature data (including mean temperature (T), mean maximum temperature (Tmax) and mean minimum temperature (Tmin)) of 15 months from September in the previous year to November in the following year during the common period of 1958–2014.

### 2.4. Study Methods

We analyzed the correlations between the chronologies of the EWW, LWW, RWW of *Pinus taiwanensis* Hayata and climate factors (T, Tmax, Tmin, P) with the Dendroclim2002 software [69] and then determined the main limiting factor on the radial growth of *Pinus taiwanensis* Hayata. Then, we tested the main limiting climate factor using the Mann–Kendell test [70] and reconstructed it with a linear regression model. In addition, we analyzed the spatial correlation between the reconstructed series and the CRU TS 4.04, 0.5° × 0.5° grid data using the KNMI climate detector, and we analyzed the periodic change by Wavelet analysis (http://climexp.knmi.nl/, accessed on 15 May 2021) and power spectrum analysis [71]. Finally, we analyzed the spatial remote correlation between the reconstructed series and the HadISST1 1° SST data in order to explore the driving mechanisms of climate change.

## 3. Results 

### 3.1. Statistical Characteristics of Different Tree Chronologies

The statistical indicators of the chronologies (Table 1) show that all indicators of RES are better than those of STD from EWW, LWW and RW. Thus, this paper chooses these RES for further research (Figure 4). The high mean sensitivity (0.231, 0.254, 0.221), SNR (15.76, 11.161, 17.348), and EPS (0.94, 0.918, 0.945) of RES from EWW, LWW and RW indicate that there may be more climate information in the chronologies. The correlation coefficients of all series and intra-tree and inter-tree indicate that the variation of earlywood width and total ring width of *Pinus taiwanensis* Hayata is consistently higher than latewood width. For these RES of EWW, LWW and RW, we selected the first year when the subsample signal intensity (SSS) [72] was more than 0.85 as the beginning of the reliable period. The reliable periods of the EWW and RW chronologies are 1901–2014. Meanwhile, the reliable period of the LWW chronology is 1904–2014.

### 3.2. The Relationships among EWW, LWW and RW Residual Chronologies

The results of the correlation between RES of EWW, LWW and RW in the common reliable period of 1904–2014 show that the correlation coefficients are 0.908 (*p* < 0.01) between RES of RW and EWW, 0.795 (*p* < 0.01) between RW and LWW, and 0.508 (*p* < 0.01) between EWW and LWW. The correlation between RES of RW and EWW is the best among the three chronologies. Combined with the statistical characteristics of the three chronologies, it can be seen that the chronologies of EWW and RW have good consistency.

### 3.3. The Correlations between RES of EWW, LWW, RW and Climatic Factors

The correlation results between RES of EWW, LWW, RW and monthly temperature and precipitation (Table 2 and Figure 5) show that there are significantly negative correlations between EWW and RW chronologies and T, Tmax, Tmin in May, June and May–June of the current year and P in October of the current year, but the significantly positive correlations between EWW and RW chronologies and P in May, June and May–June of the year. Likewise, the significantly positive correlation is between the LWW chronology and P in current May–June. In conclusion, the responses of EWW of *Pinus taiwanensis* Hayata in the Tongbai Mountains are stronger than LWW and RW versus the climate factors that are significantly negatively correlated with temperature and significantly positively correlated with precipitation in May, June and May–June of the current year. However, LWW is stronger response to the precipitation than to temperature. Obviously, the limiting factor of growth of *Pinus taiwanensis* Hayata in the Tongbai Mountains is the combination of water and heat in current May–June.

### 3.4. The Effects of Temperature before Abrupt Change on Tree Radial Growth

To understand the variation of the May–June mean temperature and mean maximum temperature (T_MJ_ and Tmax_MJ_) at the Tongbai meteorological station from 1958–2014, this paper tested them using the Mann–Kendell tests [70]. The results (Figure 6) show that both T_MJ_ and Tmax_MJ_ mutated in 2006.

Here, we performed a piecewise correlation analysis. The correlation results between the EWW chronology with T_MJ_ and Tmax_MJ_ during the pre-mutation period (1958–2005) show that there are still significantly negative correlations with −0.669 and −0.657 (*p* < 0.05), respectively. Clearly, the correlation values during the pre-mutation period (1958–2005) have increased with correlation coefficients (EW_TMJ_ = −0.598, EW_TmaxMJ_ = −0.607, *p* < 0.05) in the whole period (1958–2014). The results indicate that tree growth of *Pinus taiwanensis* Hayata has a better response to T_MJ_ during the pre-mutation period.

### 3.5. Reconstruction of T_MJ_ in the Tongbai Mountains

Considering the above results, the T_MJ_ from 1901–2005 was reconstructed by linear regression analysis based on the EWW residual chronology (RES) of *Pinus taiwanensis* Hayata in the Tongbai Mountains and T_MJ_ of the Tongbai meteorological station, and the conversion equation is as follows:$$T_{MJ}=-2.931\times W_{t}+26.028$$

In the equation, $T_{MJ}$ is the reconstructed May–June mean temperature, and $W_{t}$ is the width index of the EWW residual chronology (RES) at year t. The correlation coefficient of the equation is 0.651 (*p* < 0.01), and the variance interpretations after adjusting the degree of freedom are 42.4% and 41.1%, respectively. The reliability of the reconstructed equation is tested by the sign test method and the subsection test method, and the results are shown in Table 2. The indexes of evaluating the reliability of the reconstruction equation mainly include the correlation coefficient (R), the determination coefficient (R^2^), the efficiency coefficient (CE) and the error reduction value (RE). The reconstructed equation has a high reliability and can be used to reconstruct climate change because RE > CE > 0 (Table 3). In addition, the results of the sign test (S) and the first-order difference sign test (S1) pass the 95% and 99% confidence level, respectively. As shown in Figure 7, comparing the reconstructed T_MJ_ curve with the observed T_MJ_ curve shows that the two curves are consistent from 1958–2005.

## 4. Discussion 

### 4.1. The Main Limiting Factors of Radial Growth of Pinus taiwanensis Hayata in the Tongbai Mountains

The Tongbai Mountains belong to a monsoon humid climate, and the precipitation is concentrated in June, July and August (Figure 3). Thus, May and June are the part of the early-tree-growth season, which includes the period of EWW growth when the rainy season starts. High temperatures in May and June can accelerate the evaporation of soil water and transpiration of plants. When trees grow under high temperatures and water deficits, the leaves’ stomatal closure will prevent the loss of moisture, which prevents carbon dioxide from entering the stoma. Therefore, the tree’s photosynthetic rate decreases and the photorespiration rate increases; the net photosynthetic production also decreases, restraining tree growth [73]. Meanwhile, there is relatively more precipitation, which will relieve the suppression of high temperature stress and thus promote tree growth.

The correlation results show that T and Tmax of current May–June have a greater impact on the EW growth of *Pinus taiwanensis* Hayata in the Tongbai Mountains compared with other climatic factors. The correlation coefficients between the chronology of EWW and T and Tmax of current May–June are −0.59 and −0.60 (*p* < 0.05) (Figure 5), respectively. The research results are consistent with Mt. Yao [30] and the Dabie Mountains [57], which show that high temperature before the rainy season restrains the tree growth of different tree species in different areas. In conclusion, T_MJ_ and Tmax_MJ_ are the important factors that restrain EW radial growth of *Pinus taiwanensis* Hayata in the Tongbai Mountains.

### 4.2. T_MJ_ Variation in the Tongbai Mountains

In order to extract the more low-frequency climate information, the 11-year moving average is used to deal with the reconstructed series and obtains the 11-year moving average series (Figure 8). As shown in Figure 8, the reconstructed T_MJ_ series indicates that the May–June temperature has fluctuated frequently in the Tongbai Mountains since 1901, with temperatures ranging from 21.79–24.48 °C with the difference between the highest and lowest temperature at 2.69 °C. The horizontal line in the middle of Figure 8 represents the mean value of the reconstructed series (mean = 23.12 °C), and the standard deviation (σ) of the reconstructed series is 0.59 °C. We define the period above the mean value in the 11-year moving average series as the warm period, and the period under the mean value in the 11-year moving average series is the cold period. Thus, the warm periods include 1901–1906, 1915–1931, 1938–1944, 1951–1955, 1961–1968, 1975–1979, 1985–1988, 1997–2001, and the cold periods include 1907–1914, 1932–1937, 1945–1950, 1969–1974, 1980–1984, 1989–1996, and 2002–2005.

Similarly, we define the year when the value is over the mean value plus one standard deviation (mean + σ) (23.71 °C) as the extremely high-temperature year, and the year when the value is less than the mean value minus one standard deviation (mean−σ) (22.53 °C) as the extremely low-temperature year. There are 10 extremely high temperature years (1901, 1905, 1917, 1918, 1929, 1938, 1953, 1958, 1961, 1986), and 15 extremely low temperature years (1902, 1908, 1914, 1934, 1936, 1954,1956,1959,1970, 1980, 1984, 1991, 1993, 1996, 2004), accounting for 9.5% and 14.3% of the total reconstructed years, respectively. In addition, the high temperatures usually lead to drought events in the extremely high-temperature years; thus, we can also verify the reliability of the reconstructed series by comparing it with the records in the literature. As recorded in “China Meteorological Disaster: Henan Volume” [74], Tongbai county suffered from drought from June to August in 1918, with a small harvest in early autumn and no harvest in late autumn, resulting in famine the next spring. In 1953, there was little rain in the spring in the Nanyang area, the spring drought in the south was more severe than the north, and the total precipitation from March to May was less than 50 mm. In 1958, the precipitation in the southern Henan and Nanyang basin was less than that in the same period, the drought intensified from late May to late June, and the precipitation was less than 5 mm in mid-June. In 1961, there was a drought in southern Henan as early as the end of spring, and the drought in Nanyang Basin also began in spring and continued into the autumn. In addition, according to the “Annals of Tongbai County” [75], there was no rain in the summer for a long time in Tongbai county, and the drought was serious in 1986.

Comparing the reconstructed T_MJ_ in this study with the May–July mean minimum temperature reconstructed [62] using the *Pinus massoniana* Lamb. in the Tongbai Mountains, the results show (Figure 9) that the reconstructed T_MJ_ in this study has a relatively synchronous trend with that in the Cai and Liu curves. Meanwhile, the two curves are consistent regarding the fluctuation in cold and warm periods. The two curves show the cold periods to be in the 1910s and 1970s–1990s and the warm periods to be in the 1900s and 1920s–1940s. However, the two curves are different in amplitude and duration in the warm and cold periods as well as the cold and warm changes during the same period, which may be related to the differences of the selected tree species and the reconstructed elements.

### 4.3. Spatial Representativeness

To understand the regional representativeness of the reconstructed series in China, this paper analyzes the spatial correlation between the May–June mean temperature of the reconstructed and observed data from the Tongbai meteorological station with CRU TS 4.05 (0.5° × 0.5°) from 1958–2005 through the KNMI climate detector. As shown in Figure 10, there is better consistency in the distribution. Moreover, the correlation is significant in central and eastern China, especially in the south of the North China Plain (r > 0.5). The results show that the reconstructed T_MJ_ of the Tongbai Mountains in this study well represents central and eastern China.

### 4.4. Characteristic of Periodic Variation of Reconstructed T_MJ_

This paper analyzes the characteristics of periodic variation of the reconstructed T_MJ_ series from 1901 to 2005 in the Tongbai Mountains using a power spectrum analytical method and wavelet analytical method through the KNMI climate detector (http://climexp.knmi.nl/, accessed on 15 May 2021). Figure 11A shows that the reconstructed T_MJ_ from 1901 to 2005 in the Tongbai Mountains exists two significant cycles with 2.07a (*p* < 0.01) and 1.96a (*p* < 0.05). In addition, the wavelet analysis (Figure 11B) shows that the cycle of 2–3a appears in 1900–1910, 1930–1940, 1950–1965, 1990–2000, the cycle of 5–7a appears in 1910–1920, and the cycle of 10a appears in 1940–1950 and 1975–1990. The above results show that the periodic variation of the reconstructed T_MJ_ series in the Tongbai Mountains is mainly a quasi-biennial cycle.

### 4.5. Mechanism Analyses of Past Temperature Change

The ocean is the largest heat storage and heat conveyor belt on the surface of the Earth [76]. The heat exchange in the ocean causes changes in atmospheric circulation and then leads to a change in climate elements such as temperature and precipitation all over the world. Previous studies showed that response signals could be found from previous sea surface temperature (SST) anomalies, which indicated that the change in SST had a lag effect on air temperature [73,77,78]. In order to further study the regional factors affecting T_MJ_ changes in the Tongbai Mountains, we analyzed the spatial correlation between the reconstructed T_MJ_ series and the HadISST1 1° SST data from the previous October to the following June from 1901–2005 (Figure 12). The results showed that the reconstructed T_MJ_ was significantly negatively correlated with SST in the tropical Western Pacific Ocean and Indian Ocean and was significantly positively correlated with SST in the subtropical Western Pacific Ocean. The vector wind field in the near surface (925 hPa) shows that there were intersects of north–south airflow during the extremely low-temperature years (Figure 13A) and the southeast airflow during the extreme high-temperature years (Figure 13B) in the Tongbai Mountains.

When the tropical Western Pacific warm pool and the Indian Ocean are abnormally warm, the convective activity intensified over the tropical Western Pacific around the Philippines [79,80,81]. Compared with normal years, the position of the Western Pacific Subtropical High (WPSH) is more northerly, the north jump is earlier, and the outbreak of the East Asian monsoon is earlier, resulting in the position of the rain belt being more northerly and moving rapidly northward. In addition, the high-latitude cold air force is strong and frequently moves southward [82,83], inducing a north–south airflow that frequently intersects in May and June in the lower troposphere (925 hPa) near the Tongbai Mountains (Figure 13A), eventually resulting in the lower May–June mean temperature in the Tongbai Mountains. 

When the tropical Western Pacific warm pool and the Indian Ocean are abnormally cold, the convection activities weaken over the tropical Western Pacific around the Philippines [79,80,81]. The position of the WPSH is more southerly, the position of the monsoon rain belt is more southerly and moves northward slowly [82,83], and the Tongbai Mountains are affected by the single southeast warm airflow (Figure 13B), resulting in the higher May–June mean temperature in the Tongbai Mountains. Some studies [84,85,86,87] have shown that SST over the tropical Western Pacific warm pool, the Indian Ocean, and the WPSH have a quasi-oscillation cycle of 2–3a, which is consistent with a significant period of the reconstructed T_MJ_ in this study. These results indicate that the reconstructed T_MJ_ is significantly positively and negatively correlated with the SST in the subtropical Western Pacific Ocean and the tropical Western Pacific warm pool and Indian Ocean, which are related to the quasi-biennial oscillation (QBO) [88,89]. QBO is defined as the quasi-biennial interannual oscillation of SST over the Indo-Pacific equatorial region. Sea–air oscillation (QBO) may affect central China through the transport of water vapor in the East Asian summer monsoon, the South China Sea summer monsoon, and the Indian summer monsoon.

## 5. Conclusions

This study established the residual chronologies of EWW, LWW and RW of *Pinus taiwanensis* Hayata, which were collected from the natural forest in the Tongbai Mountains. We found that EWW residual chronology has more climate information and better response to the climate factors than LWW and RW through statistical characteristics of the chronologies and correlations between the chronologies and the climate factors. It was also found that the limiting factors that restrained EW radial growth of *Pinus taiwanensis* Hayata in the Tongbai Mountains were May–June mean temperature and May–June mean maximum temperature of the current year. The correlation between EWW and mean temperature in May–June in the pre-mutation period (1958–2005) is significantly negative (R = −0.669, *p* < 0.05), and the correlation was stable, which could be used for reconstruction.

Therefore, this paper reconstructed the T_MJ_ in the Tongbai Mountains from 1901 to 2005 and processed the reconstructed series using the 11-year moving average method. The variance of the reconstruction equation was 42.4% (41.1% after adjusting for the degrees of freedom), and the reconstructed T_MJ_ in the Tongbai Mountains experienced eight warm periods (1901–1906, 1915–1931, 1938–1944, 1951–1955, 1961–1968, 1975–1979, 1985–1988, 1997–2001) and eight cold periods (1907–1914, 1932–1937, 1945–1950, 1969–1974, 1980–1984, 1989–1996, 2002–2005). The reconstructed T_MJ_ in this study has a relatively synchronous trend with that of Cai et al., and the two curves are consistent regarding the fluctuations in cold and warm periods. Meanwhile, the spectral analysis and the wavelet analysis found that the reconstructed T_MJ_ in the Tongbai Mountains had a significant cycle of 2a that appeared the most over the past 105 years. The spatial correlation analysis indicated that the reconstructed T_MJ_ series can well represent central and eastern China. A significantly negative correlation with SST over the tropical Western Pacific Ocean and Indian Ocean and a significantly positive correlation with SST over the subtropical Pacific Ocean from the previous October to the current June indicated that the T_MJ_ periodic fluctuations in the Tongbai Mountains might be related to the quasi-biennial interannual oscillation of SST over the Indo-Pacific equatorial region (QBO). The results of this study are significant for further understanding and exploration of forest growth and climate change in the climatic transition zone.

## Figures and Tables

**Figure 1 biology-11-01077-f001:**
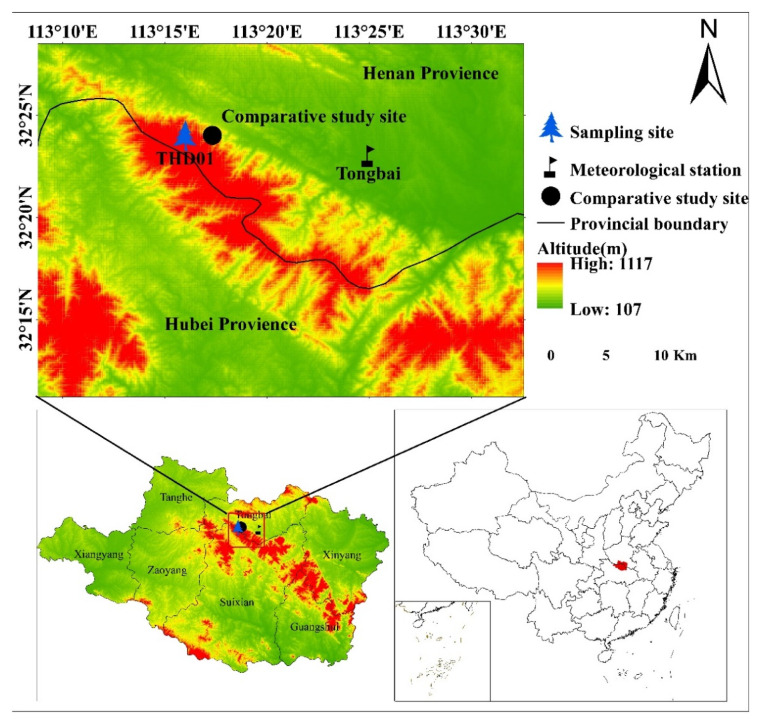
Location map of sampling site, comparative study site and nearby meteorological stations.

**Figure 2 biology-11-01077-f002:**
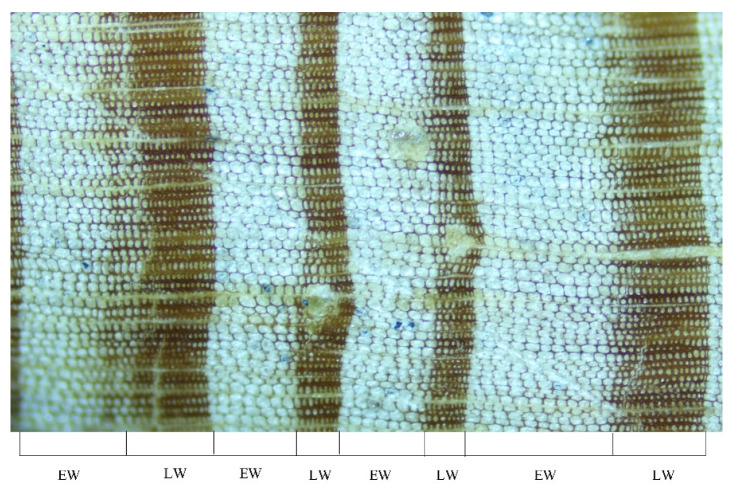
The boundary diagram of EW and LW of *Pinus taiwanensis* Hayata under a microscope, magnification of 200 times.

**Figure 3 biology-11-01077-f003:**
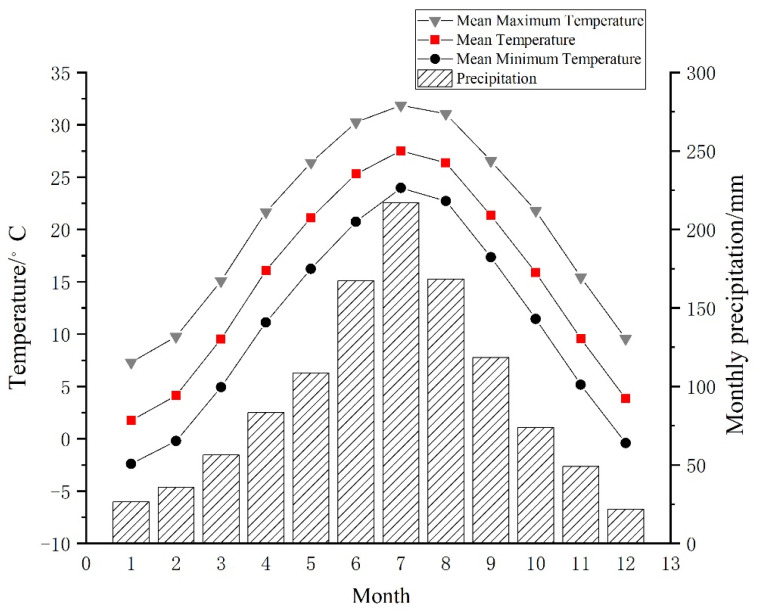
The distribution graph of monthly mean temperature (T), mean maximum temperature (Tmax), mean minimum temperature (Tmin) and monthly total precipitation (P) at the Tongbai meteorological station from 1958–2014.

**Figure 4 biology-11-01077-f004:**
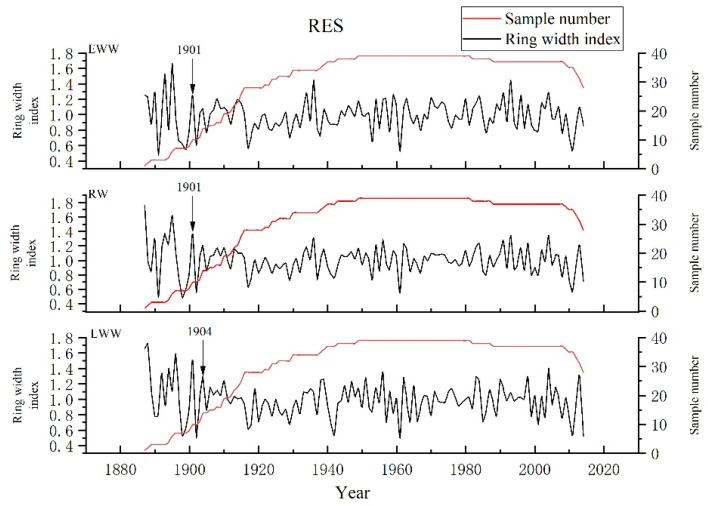
RES of EWW, LWW and RW and the samples of *Pinus taiwanensis* Hayata in the Tongbai Mountains. (Arrow shows the beginning year of SSS > 0.85).

**Figure 5 biology-11-01077-f005:**
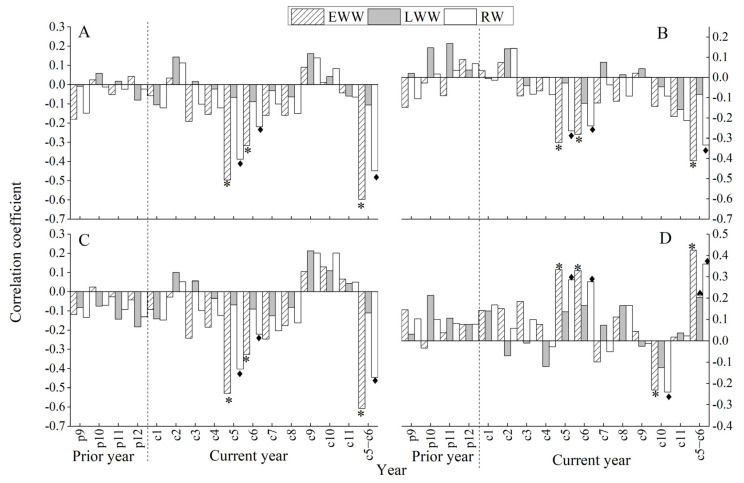
The correlation analyses between RES of EWW, LWW, RW and T (**A**), Tmin (**B**), Tmax (**C**), P (**D**) of the Tongbai meteorological station. *, and ◆ all stand for *p* < 0.05.

**Figure 6 biology-11-01077-f006:**
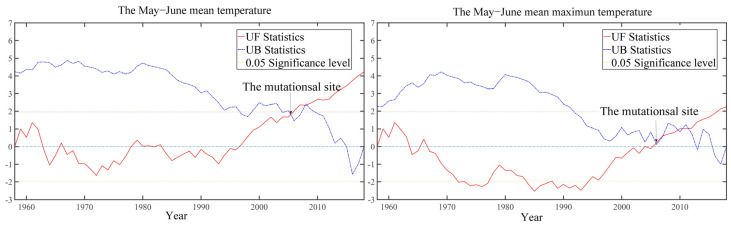
Mann–Kendell tests of T_MJ_ and Tmax_MJ_ at the Tongbai meteorological station from 1958–2014.

**Figure 7 biology-11-01077-f007:**
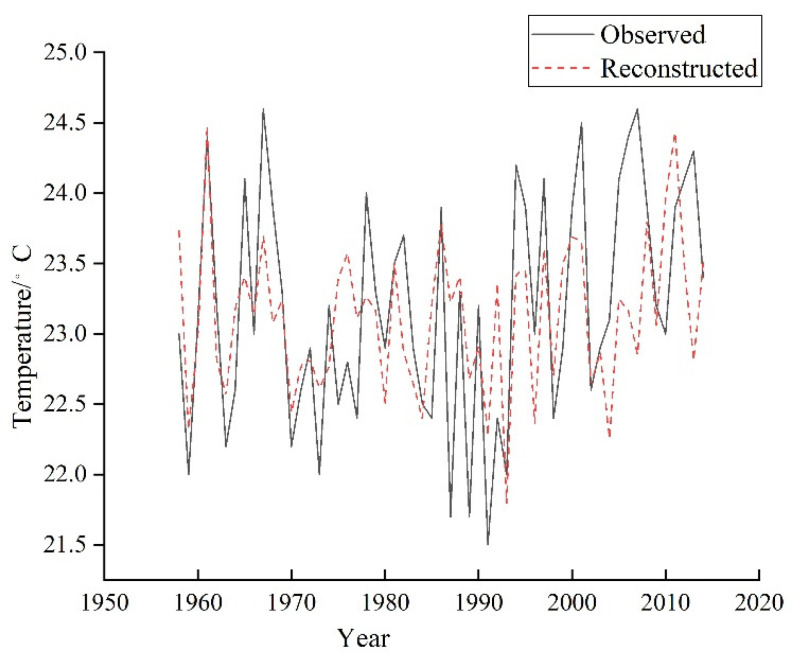
Comparison of the reconstructed and observed T_MJ_ in the Tongbai Mountains from 1958 to 2018.

**Figure 8 biology-11-01077-f008:**
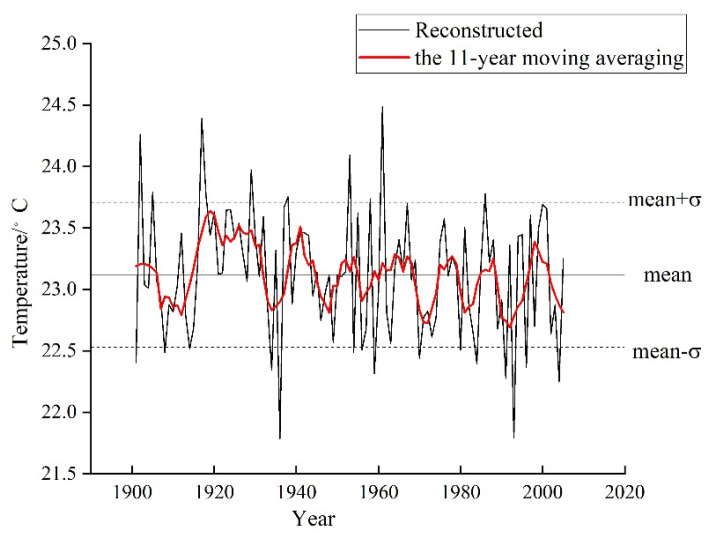
The reconstructed T_MJ_ curve and 11-year moving average curve in the Tongbai Mountains from 1901 to 2005.

**Figure 9 biology-11-01077-f009:**
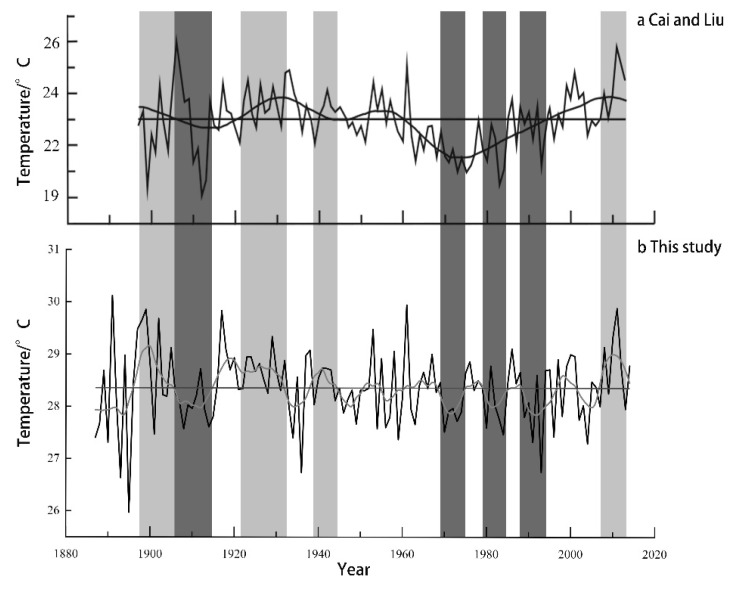
The comparison of the moving average curves of the reconstructed T_MJ_ in this study (**b**) and the reconstructed May–July mean minimum temperature of Cai and Liu (**a**) in the Tongbai Mountains. Dark colors and light colors represent warm periods and cold periods, respectively.

**Figure 10 biology-11-01077-f010:**
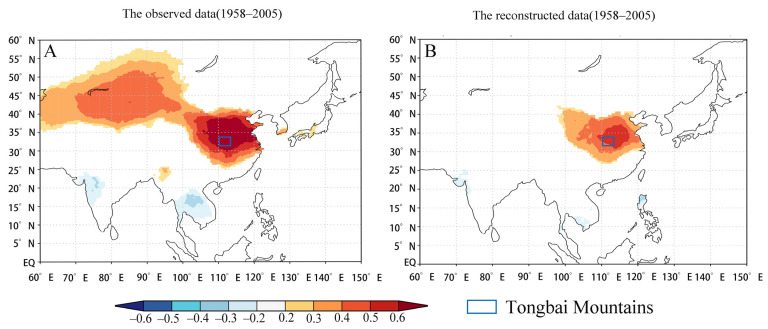
Spatial correlation between the observed (**A**) and reconstructed (**B**) T_MJ_ with CRU TS 4.05 (0.5° × 0.5°) from 1958–2005 through the KNMI climate detector, which was used in order to study the regional representativeness of the reconstructed series in China.

**Figure 11 biology-11-01077-f011:**
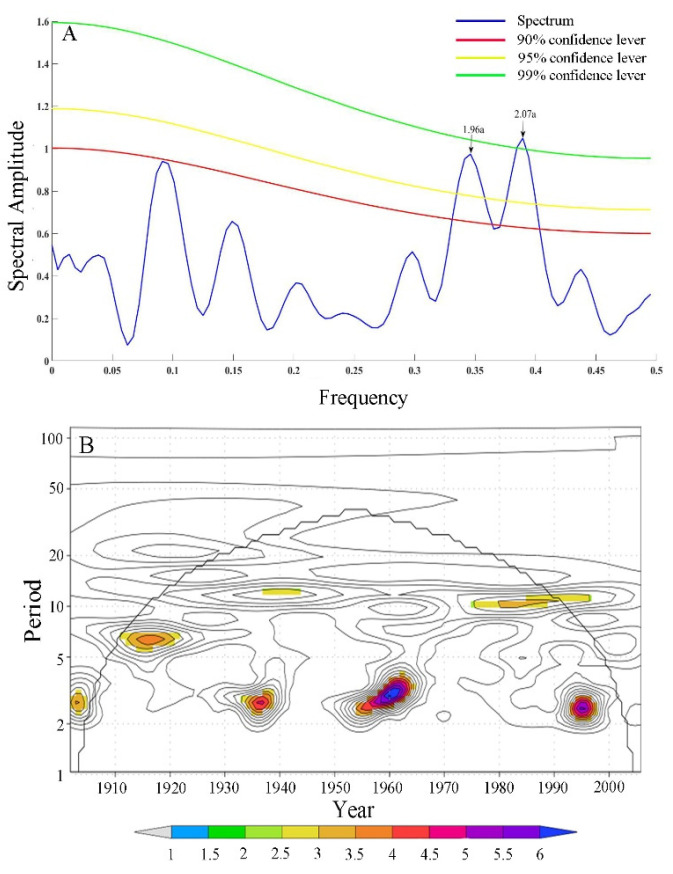
The periodic variation of the reconstructed T_MJ_ series from 1901 to 2005 in the Tongbai Mountains using the power spectrum analysis (**A**) and wavelet analysis (**B**); 2.07a and 1.96a (**A**) are significant cycles of the reconstructed T_MJ_ from 1901 to 2005 in the Tongbai Mountains, which reach 99% and 95% confidence levels, respectively.

**Figure 12 biology-11-01077-f012:**
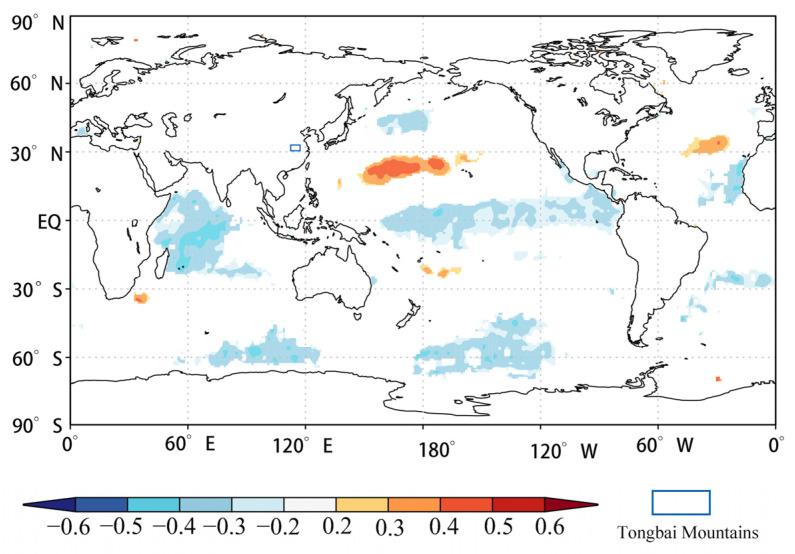
The spatial teleconnection between the reconstructed T_MJ_ series and the HadISST1 1° sea surface temperature during the common period of 1901–2005.

**Figure 13 biology-11-01077-f013:**
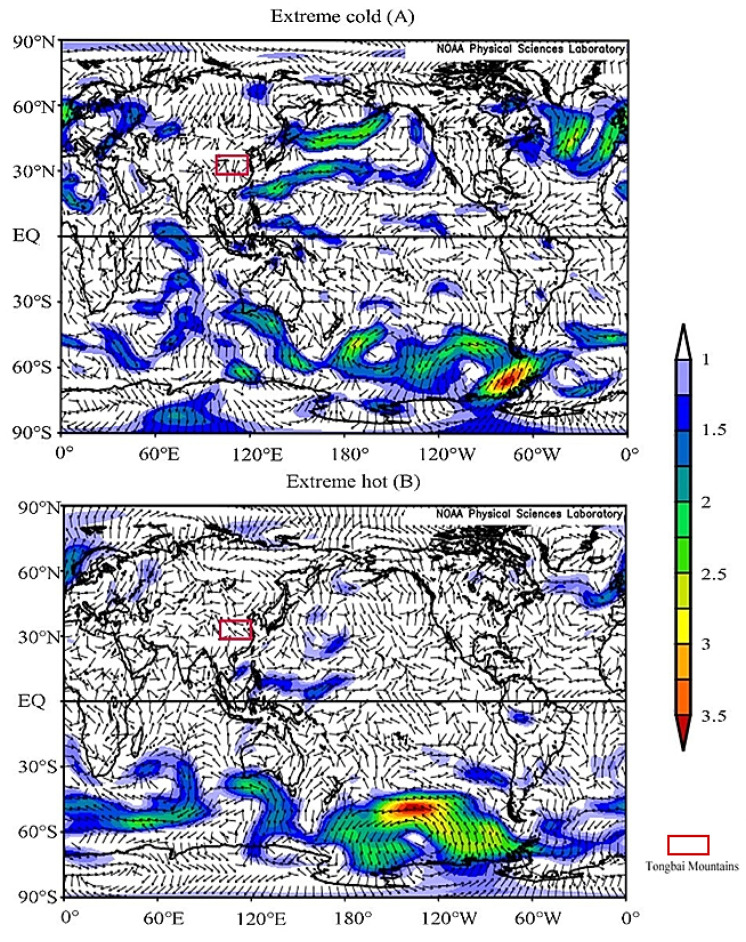
The changes in vector wind field during the May–June mean temperature in the Tongbai Mountain area in extremely hot years (**A**) and extremely cold years (**B**).

**Table 1 biology-11-01077-t001:** The chronological statistics of EWW, LWW and RW of *Pinus taiwanensis* Hayata.

Statistical Indicator	EWW		LWW		RW	
	RES	STD	RES	STD	RES	STD
Core/tree	39/20	39/20	39/20	39/20	39/20	39/20
Common period	1887–2014	1887–2014	1887–2014	1887–2014	1887–2014	1887–2014
Mean sensitivity	0.231	0.226	0.254	0.237	0.221	0.198
All series correlation coefficient	0.314	0.249	0.258	0.199	0.34	0.239
Intra-tree correlation coefficient	0.541	0.53	0.475	0.423	0.56	0.508
Inter-tree correlation coefficient	0.308	0.241	0.253	0.193	0.334	0.231
Signal noise ratio	15.76	11.681	11.161	7.437	17.348	9.595
Expressed population signal	0.94	0.921	0.918	0.881	0.945	0.906

**Table 2 biology-11-01077-t002:** The significant correlation analyses between RES of EWW, LWW, RW and T, Tmin, Tmax, P of the Tongbai meteorological station (*p* < 0.05).

Month	EWW				RW				LWW
	T	Tmin	Tmax	P	T	Tmin	Tmax	P	P
Current May	−0.496	−0.32	−0.528	0.334	−0.388	−0.264	−0.402	0.285	
Current June	−0.316	−0.281	−0.326	0.329	−0.219	−0.239	−0.22	0.278	
Current October				−0.23				−0.239	
Current May–June	−0.598	−0.411	−0.607	0.425	−0.449	−0.334	−0.446	0.36	0.203

**Table 3 biology-11-01077-t003:** Test statistical results of the reconstruction equation.

	Calibration (1958–1987)	Verification (1988–2005)	Calibration (1976–2005)	Verification (1958–1975)	Full Calibration (1958–2005)
R	0.603	0.726	0.596	0.747	0.651
R^2^	0.363	0.527	0.355	0.559	0.424
RE		0.508		0.552	
CE		0.505		0.551	
S	18+/12−	14+/4− *	21+/9− *	11+/7−	32+/16− *
S1	3 **	3 **	5 **	1 **	12 **

* represents a 95% confidence level; ** represents a 99% confidence level; + represents the same number; − represents the different number.

## Data Availability

Not applicable.

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
