# Peer review of "Early Summer Temperature Variation Recorded by Earlywood Width in the Northern Boundary of Pinus taiwanensis Hayata in Central China and Its Linkages to the Indian and Pacific Oceans"

_biology, 2022, doi:10.3390/biology11071077_

Round 1

Reviewer 1 Report

Peng et al. contribute a nice study of climate reconstruction in central China, where such climate proxies are still scarce. The methods used are regular and sound and the main results obtained are generally supportive for the main conclusions. A suit of pretty figures was generated and well organized in the manuscript. I would be happy to see the publication of this study after a minor revision.

Specific points:

L190: High mean sensitivity may be not an indicator of climate information.

L254: Awkward sentence.

Fig.9: I suppose the lines are the multi-year mean values but you should have clearly stated here.

Fig. 10: The hollow parts should also be explained in the figure caption (why and how was masked out).

L358: Awkward sentence. Besides, more information should be provided on the meanings of the symbols in the figure.

L397: “correlated with”

Fig.12: The period over which the correlations were calculated should be provided.

Author Response

Author's Reply to the Review Report (Reviewer 1):

Thank Reviewer very much for your comments and suggestions. We modified the following questions point to point.

L190: High mean sensitivity may be not an indicator of climate information.

Author's ReplyThanks very much! “High mean sensitivity may be not an indicator of climate information. ”is correct. However, the study area is a climatic transitional section and climate change has a great influence on tree growth, so high mean sensitivity is an indicator of climate information.

L254: Awkward sentence.

Author's ReplyThanks very much! We have changed “To understand variation of TMJ and TmaxMJ during 1958-2014yr, Mann-Kendell tests [70] were performed to detect if their changes are abnormal. The results (Figure 6) found that TMJ and TmaxMJ both abrupt shifts in 2006yr.”into “To understand variation of TMJ and TmaxMJ during 1958-2014yr, this paper tested them by the Mann-Kendell tests [70]. The results showed that both TMJ and TmaxMJ mutated in 2006yr.”

Fig.9: I suppose the lines are the multi-year mean values but you should have clearly stated here.

Author's ReplyThanks very much! We have added on the figure caption “The comparison of the moving average curves of the reconstructed TMJ in this study (b) and the reconstructed May-July mean minimum temperature of Cai and Liu (a) in Tongbai Mountains.”.

Fig. 10: The hollow parts should also be explained in the figure caption (why and how was masked out).

Author's ReplyThanks very much! We have added in the figure caption “Spatial correlation between observed (A) and reconstructed (B) TMJ with CRU TS 4.05 (0.5°×0.5°) during 1958-2005 through the KNMI climate detector, in order to study the regional representativeness of the reconstructed series in China.”.

L358: Awkward sentence. Besides, more information should be provided on the meanings of the symbols in the figure.

Author's ReplyThanks very much! We have changed the figure caption “The power spectrum analysis (A) wavelet analysis (B) and of the reconstructed TMJ series in Tongbai Mountains.” into “The periodic variation of the reconstructed TMJ series from 1901 to 2005 in Tongbai Mountains by the power spectrum analysis (A) and the wavelet analysis (B). 2.07a and 1.96a (A) are significant periods of the reconstructed TMJ from 1901 to 2005 in Tongbai Mountains which reach 99% and 95% confidence levels, respectively.”.

L397: “correlated with”.

Author's ReplyThanks very much! We have changed “the reconstructed TMJ is significantly positive/negative correlation with the SST” into“the reconstructed TMJ is significantly positive/negative correlated with the SST”.

Fig.12: The period over which the correlations were calculated should be provided.

Author's ReplyThanks very much! We have revised the figure caption “The spatial teleconnection between the reconstructed TMJ series with the HadISST1 1° sea surface temperature.” as “The spatial teleconnection between the reconstructed TMJ series with the HadISST1 1° sea surface temperature during the common period of 1901-2005.”.

Reviewer 2 Report

Dear Authors,

thank you for a presented draft of your manuscript.

I would like to share with you my  comments and suggestions for presented research paper (numbered by row Nr.).

79  shorcut PDSI index could be shortly described

93-95  isn´t necesarry to repeat in every sentence scientific name Pinus Taiwanensis Hayata (also check please correct scientific name)

126-134  please clear formulate this paragraph. Do you mean natural association?

144-151  please unify you writing style in passive (check whole draft)

154  "above mentined criteria"

163  it woul be nice to shortly explain what Standard chronology and Residual chronology means

172  describe the way how and why you chose last 15 months for a computation of trends? You make a trend for every singel year in the period 1887-2014?

177-186  passive

194 "better consistent" other clear formulation please

198-200 How many samples you finaly used? Only 14 in total? Or 14 for whole period? Which period you mean (1904-2014 or 1901-2014)? Please explain clearly.

Tab.2  Pleas comment shortcuts used in table. Why are you speaking only about current May-June and not about upper mentioned periods? Add some explanation to Material and methods (you can use parts from Discusion).

229-247  Some parts would be better shift to Discusion.

252  The period 1958-2014 - what is this for period. Selected by, why? Base on what?

Fig.7 is it just demonstration of how well fitted is your model? You already have date from meteo stations from the region.

336  Please what exately showing the map? Is it natural extensin of P. taiwanensis or potentional area of appearence of P. taiwanensis by reconstructed climate. You fitted you model by date from local wether station, yes? So is not a suprice that you have a high correlation there.

351 explain please what values 2.07a and 1.96a means

379-401  Whole paragraph is only  summary of prewious works. I don´t see here any original results which you are comented. Positive/negative correlation? Wchich correlation you obtained with your data?

423-428  QBO oscilations - please comment wchich kind of oscilation.

Dear authors,

I would appreciate to see have more axact described part of Material and methods especialy some data about age of tested trees,  Nr. of samples, why you reduced the periods (in draft appearing many different periods without explanation why and base on what they were chosen). Please check entire style of your manuscrip.

Thank you.

Author Response

Author's Reply to the Review Report (Reviewer 2):

Thank Reviewer very much for your comments and suggestions. We modified the following questions point to point.

Line 79 shortcut PDSI index could be shortly described

Author's ReplyThanks very much! We have revised “scPDSI index” as “self-calibrated Palmer Drought Severity Index”.

Line 93-95 isn´t necessary to repeat in every sentence scientific name Pinus Taiwanensis Hayata (also check please correct scientific name)

Author's ReplyThanks very much! We have revised “the more tree-ring researches in the Dabie Mountains were achieved, mainly including dendroecology of single tree species such as P. Taiwanese’s Hayata[53-56] and different tree species such as P. Tabulaeformis and P. Taiwanese’s Hayata[57], P. massoniana and P. Taiwanese’s Hayata[56] and dendroclimatology based on different tree species such as P. Taiwanese’s Hayata[58-60] and P. massoniana[61].” as “the more tree-ring researches in the Dabie Mountains were achieved, mainly including dendroecology and dendroclimatology based on Pinus taiwanensis Hayata and Pinus tabulaeformis Carr. as well as Pinus massoniana Lamb. [53-61].”.

Line 126-134 please clear formulate this paragraph. Do you mean natural association?

Author's ReplyThanks very much! In this paragraph, we want to introduce the geographical location, the characteristics of climate and soil as well as vegetation in Tongbai Mountains. Then, we want to explain the necessity of this study through the advantages of the study area and Pinus taiwanensis Hayata. We have revised this paragraph as “The vegetation in Tongbai Mountains is characterized by a north-south transition, which is a mixed evergreen coniferous broad-leaved forest and deciduous broad-leaved forest. There are many types of vegetation in Tongbai Mountains, such as Pine forests、Quercus forests and bushes, includes Pinus taiwanensis Hayata、Pinus massoniana Lamb.、Pinus tabulaeformis Carr.、Quercus variabilis Blume、Quercus glauca Thunb.、Quercus acutissima Carruth.、Cotinus coggygria Scop.、Forsythia suspensa (Thunb.) Vahl、Rhododendron simsii Planch etc. [64]. Because Tongbai Mountains is located in the climate transition zone and the northern and western distribution boundary of Pinus taiwanensis Hayata in mainland China [55], the natural forest of Pinus taiwanensis Hayata in Tongbai Mountains is an ideal tree species to study the relationship between tree-ring and climate factors.”.

Line 144-151 please unify you writing style in passive (check whole draft)

Author's ReplyThanks very much! We have revised as “The boundary of EW and LW was clear and easy to distinguish in most cases, but it was difficult to distinguish the boundary of the transition between EW and LW in a few sample cores, so this paper determined the standard boundary of EW and LW based on the method of determining of Stahle et al. [43]. As shown in Figure 2 which was captured by an Olympus CX 43 light microscope (200× magnification), (1) we defined the mutant line as the boundary between EW and LW where was a mutation from big-light cells to small-dark cells. (2) We defined the middle line of the gradual change area as the boundary where was a gradual change between EW and LW.”.

Line 154 "above mentioned criteria"

Author's ReplyThanks very much! We have revised “the above mentioned criteria” as “the above-mentioned criteria”.

Line 163 it would be nice to shortly explain what Standard chronology and Residual chronology means

Author's ReplyThanks very much! We have added “The standard chronology removes the trees age-related growth trend, while the residual chronology removes not only the trees age-related growth trend but also the persistent influence of individual trees on later growth due to local microenvironmental changes.”.

Line 172 describe the way how and why you chose last 15 months for a computation of trends? You make a trend for every single year in the period 1887-2014?

Author's ReplyConsidering the climatic factors in the previous late growing season affecting the tree growth in the following year, we select climate factors of the last 15 months from September in the previous year to November in the current year during the common period of 1958-2014. We analyze the correlation between the chronologies of the EWW, LWW, RWW of Pinus taiwanensis Hayata and climate factors during the common period of 1958-2014.

Line 177-186 passive

Author's ReplyThanks very much! We have revised as “This paper analyzed the correlation between the chronologies of the EWW, LWW, RWW of Pinus taiwanensis Hayata and climate factors (T, Tmax, Tmin, P) by Dendroclim2002 software [69], then determined main limiting factor on the radial growth of Pinus taiwanensis Hayata. Then, we tested the main limiting climate factor by Mann-Kendell test [70] and reconstructed it by the linear regression model. In addition, we analyzed the spatial correlation between reconstructed series and CRU TS 4.04, 0.5°×0.5° grid data by the KNMI climate detector and the periodic change by Wavelet analysis (http://climexp.knmi.nl/) and power spectrum analysis [71]. Finally, we analyzed the spatial remote correlation between reconstructed series and HadISST1 1° SST data in order to explore the driving mechanism of climate change.”.

Line 194 "better consistent" other clear formulation please

Author's ReplyThanks very much! We have revised “better consistent” as “high consistent”.

Line 198-200 How many samples you finally used? Only 14 in total? Or 14 for whole period? Which period you mean (1904-2014 or 1901-2014)? Please explain clearly.

Author's ReplyWe finally used 20 trees and 39 cores. We select the first year when the subsample signal intensity (SSS) is more than 0.85 as the beginning of reliable period. The reliable periods of EWW and RW chronologies are 1901-2014. Meanwhile, the reliable time period of LWW chronology is 1904-2014.

Tab.2 Please comment shortcuts used in table. Why are you speaking only about current May-June and not about upper mentioned periods? Add some explanation to Material and methods (you can use parts from Discussion).

Author's Reply The figure 5 shows the correlation between RES of EWW, LWW and RW with climate factors of 15 months from September in the previous year to November in the current year. The table 2 only shows the significant correlation between RES of EWW, LWW and RW with climate factors.

Line 229-247 Some parts would be better shift to Discussion.

Author's ReplyThanks very much! We have changed this part into Discussion and separated the Results from the Discussion.

Line 252 The period 1958-2014 - what is this for period. Selected by, why? Based on what?

Author's Reply The meteorological record period of Tongbai meteorological station is 1958-2018. But we collected the tree-ring in 2015, so we selected the meteorological record during 1958-2014.

Fig.7 is it just demonstration of how well fitted is your model? You already have date from meteo stations from the region.

Author's Reply We want to demonstrate the reconstructed data is reliable by comparing the reconstructed and observed data during 1958-2005.

Line 336 Please what exactly showing the map? Is it natural extensin of P. taiwanensis or potentional area of appearence of P. taiwanensis by reconstructed climate? You fitted you model by date from local wether station, yes? So is not a suprice that you have a high correlation there.

Author's ReplyThanks very much! In order to study the regional representativeness of the reconstructed series in China, we analyze the spatial correlation between the reconstructed and observed temperature data with the CRU TS 4.05 (0.5°×0.5°) temperature data during 1958-2005 through the KNMI climate detector, respectively. We want to show that the reconstructed data is reliable by the figure 10.

Line 351 explain please what values 2.07a and 1.96a means

Author's ReplyThanks very much! The 2.07a and 1.96a are cycles of 2.07 and 1.96 years.

Line 379-401 Whole paragraph is only summary of previous works. I don´t see here any original results which you are commented. Positive/negative correlation? Which correlation you obtained with your data?

Author's ReplyPrevious studies showed that the change of sea surface temperature has a lag effect on the air temperature. We analyzed the spatial correlation between reconstructed TMJ series and the HadISST1 1° SST data from the previous October to the current June during 1901-2005. The results showed that the reconstructed TMJ was significantly negative correlated with SST in the Tropical Western Pacific Ocean and Indian Ocean and significantly positive correlated with SST in the Subtropical Western Pacific Ocean. In order to explain this result, we obtain the possible driving mechanism by reviewing relevant literatures and verify our conclusion by the vector wind field.

Line 423-428 QBO oscillations - please comment which kind of oscillation.

Author's ReplyThanks very much! We have added “QBO means the quasi-biennial interannual oscillation of SST over the Indo-Pacific equatorial region”

Reviewer 3 Report

This is a very interesting paper correlating growth ring characteristics of Pinus taiwanensis with different climatic periods and their atmospheric characteristics, including records of seawater temperature oscillations in areas of the Pacific Ocean near China.

I found the paper to be well structured, although some of the points I indicate by line number should be corrected:

Title: Please, change the name of the species: Pinus taiwanensis Hayata.

Species names in any biology paper must be spelled correctly. On the other hand, I have found Pinus taiwanensis presents a certain variability with the variety Pinus taiwanensis var. damingshanensis W.C.Cheng & L.K.Fu from China. What taxa are you treating? Perhaps, you treat the species sensu lato, but this need to be indicated as Pinus taiwanensis s.l.

Introduction

Line 67: Please, include for Douglas fir and Jack Pine the scientific names at least, the first time appearing in the text.

Line 74: Please, plant names must be spelled correctly according to the International Plant Names Index: Picea meyeri Rehder & E.H.Wilson; Picea Mayeri does not exist.

Line 75: Cryptomeria fortune? Its name is Cryptomeria fortunei Hooibr. ex Billain, but this name is a synonym of Cryptomeria japonica (Thunb. ex L.f.) D.Don. This latter is the correct name.

Line 77: Please, write the name correctly: Pinus tabuliformis Carrière.

Line 80: Please, write the name correctly: Pinus armandi Franch.

Line 84: Which Pinus species?: P. massoniana Lamb. or P. massoniana Siebold & Zucc.? This latter is a synonym of P. thunbergii Parl. and P. thunbergii is the correct name.

Line 84: Which is the genus ‘T’?. I don't know what is ‘T’.

Line 88: Please, write the name correctly: Pinus taiwanensis.

All these nomenclatural and taxonomic aspects should be observed throughout the text.

Study area

In this paragraph should be a reference to the figure 1, indicating the study area.

Lines 125 and 126: The sentence ‘The original forest is seriously damaged by human factors, and only a few natural forest areas are not disturbed by human factors, such as P. Taiwanense’s Hayata, P. massoniana and Cork oak’ is incorrectly formulate. I don´t find its sense and also the words ‘by human factors’ is repeated.

Line 127 to 130:  Authors have written species but not vegetation types: Pine forests, Quercus forests or indicate a vegetation composed by the following species. You cannot say that a vegetation type is a species list. Vegetation and flora are different concepts.

Please, write the species names correctly, and their authorities: Quercus variabilis Blume, Rhododendron simsii Planch. Please check the others.

2.3. Climate data

Where were the meteorological data taken from?

Figures

The figure 2 should include in the caption the optical method you have used to show the cells.

Figure 4: What do the black and red lines mean? indicate it in the caption.

Figure 9: I think the figure should first draw and comment on Cai & Liu's work (a) and then that of the authors (b), which is later (b).

References

Line 578: Change ‘Gengraphy’ to ‘Geography’

In essence, as a reviewer, I seek greater rigour in the botanical aspects of the work, even in general is very interesting.

Author Response

Author's Reply to the Review Report (Reviewer 3):

Thank Reviewer very much for your comments and suggestions. We modified the following questions point to point.

Title: Please, change the name of the species: Pinus taiwanensis Hayata.

Author's ReplyThanks very much! We have revised “May-June temperature variation recorded by earlywood width in the northern boundary of Pinus Taiwanese’s Hayata, central China and its linkages to the Indian and Pacific Oceans” as “May-June temperature variation recorded by earlywood width in the northern boundary of Pinus taiwanensis Hayata, central China and its linkages to the Indian and Pacific Oceans”.

Line 67: Please, include for Douglas fir and Jack Pine the scientific names at least, the first time appearing in the text.

Author's ReplyThanks very much! We have revised “Douglas fir and Jack Pine” as “Pseudotsuga menziesii” and “Pinus ponderosae”.

Line 74: Please, plant names must be spelled correctly according to the International Plant Names Index: Picea meyeri Rehder & E.H.Wilson; Picea Mayeri does not exist.

Author's ReplyThanks very much! We have revised it as “Picea meyeri Rehd. et Wils.”.

Line 75: Cryptomeria fortune? Its name is Cryptomeria fortunei Hooibr. ex Billain, but this name is a synonym of Cryptomeria japonica (Thunb. ex L.f.) D.Don. This latter is the correct name. Author's ReplyThanks very much! We have revised it as “Cryptomeria japonica (L. f.) D. Don”.

Line 77: Please, write the name correctly: Pinus tabuliformis Carrière.

Line 80: Please, write the name correctly: Pinus armandi Franch.

Author's ReplyThanks very much! We have revised “Pinus tabulaeformis and Pinus armandii” as “Pinus tabulaeformis Carr.” and “Pinus armandi Franch.”.

Line 84: Which Pinus species?: P. massoniana Lamb. or P. massoniana Siebold & Zucc.? This latter is a synonym of P. thunbergii Parl. and P. thunbergii is the correct name.

Author's ReplyThanks very much! The specie is “Pinus massoniana Lamb.” and we have revised it.

Line 84: Which is the genus ‘T’? I don't know what is ‘T’.

Author's ReplyThis specie is “Tsuga longibracteata Cheng” and we have revised it.

Line 88: Please, write the name correctly: Pinus taiwanensis.

Author's ReplyWe have revised “P. Taiwanense’s Hayata” as “Pinus taiwanensis Hayata”.

Lines 125 and 126: The sentence ‘The original forest is seriously damaged by human factors, and only a few natural forest areas are not disturbed by human factors, such as P. Taiwanense’s Hayata, P. massoniana and Cork oak’ is incorrectly formulate. I don´t find its sense and also the words ‘by human factors’ is repeated.

Author's ReplyWe have deleted it.

Line 127 to 130:  Authors have written species but not vegetation types: Pine forests, Quercus forests or indicate a vegetation composed by the following species. You cannot say that a vegetation type is a species list. Vegetation and flora are different concepts. Please, write the species names correctly, and their authorities: Quercus variabilis Blume, Rhododendron simsii Planch. Please check the others.

Author's ReplyThanks very much! We have described as “There are many types of vegetation in Tongbai Mountains, such as Pine forests、Quercus forests and bushes, includes Pinus taiwanensis Hayata、Pinus massoniana Lamb.、Pinus tabulaeformis Carr.、Quercus variabilis Blume、Quercus glauca Thunb.、Quercus acutissima Carruth.、Cotinus coggygria Scop.、Forsythia suspensa (Thunb.) Vahl、Rhododendron simsii Planch etc.”.

2.3. Climate data: Where were the meteorological data taken from?

Author's ReplyThanks very much! The meteorological data were selected from China Meteorological Data Service Centre (http://data.cma.cn/).

The figure 2 should include in the caption the optical method you have used to show the cells. Author's ReplyThanks very much! The figure 2 is captured by an Olympus CX 43 light microscope (200× magnification).

Figure 4: What do the black and red lines mean? indicate it in the caption.

Author's ReplyThanks very much! The red lines mean the sample number and the black lines mean ring width index. We have revised in the figure 4.

Figure 9: I think the figure should first draw and comment on Cai & Liu's work (a) and then that of the authors (b), which is later (b).

 Author's ReplyThanks very much! We have revised in the figure 9.

Line 578: Change ‘Gengraphy’ to ‘Geography’.

Author's ReplyThanks very much! We have revised “Gengraphy” as “Geography”.

Round 2

Reviewer 2 Report

Dear authors,

thank you for all explanations and corrections you did in presented paper. I hope it helps to better understanding of presented research.

I am satisfied and don´t have any other question.

Wish you good luck and success in your futere research.

Best Regards

Author Response

Dear reviewer,

Thank you very much for your comments and suggestions. Those comments and suggestions are all valuable and very helpful for revising and improving our paper, as well as the important guiding significance to our researches. On behalf of my co-authors, we would like to express our great appreciation to you.

Best regards!

Sincerely yours,

Meng Peng

College of Geography and Environmental Science, Henan University